# Machine Learning Models for the Prediction of Renal Failure in Chronic Kidney Disease: A Retrospective Cohort Study

**DOI:** 10.3390/diagnostics12102454

**Published:** 2022-10-11

**Authors:** Chuan-Tsung Su, Yi-Ping Chang, Yuh-Ting Ku, Chih-Ming Lin

**Affiliations:** 1Department of Healthcare Information and Management, Ming Chuan University, Taoyuan 333, Taiwan; 2Department of Nephrology, Taoyuan Branch of Taipei Veterans General Hospital, Taoyuan 330, Taiwan

**Keywords:** kidney disease, renal dialysis, random forest

## Abstract

This study assessed the feasibility of five separate machine learning (ML) classifiers for predicting disease progression in patients with pre-dialysis chronic kidney disease (CKD). The study enrolled 858 patients with CKD treated at a veteran’s hospital in Taiwan. After classification into early and advanced stages, patient demographics and laboratory data were processed and used to predict progression to renal failure and important features for optimal prediction were identified. The random forest (RF) classifier with synthetic minority over-sampling technique (SMOTE) had the best predictive performances among patients with early-stage CKD who progressed within 3 and 5 years and among patients with advanced-stage CKD who progressed within 1 and 3 years. Important features identified for predicting progression from early- and advanced-stage CKD were urine creatinine and serum creatinine levels, respectively. The RF classifier demonstrated the optimal performance, with an area under the receiver operating characteristic curve values of 0.96 for predicting progression within 5 years in patients with early-stage CKD and 0.97 for predicting progression within 1 year in patients with advanced-stage CKD. The proposed method resulted in the optimal prediction of CKD progression, especially within 1 year of advanced-stage CKD. These results will be useful for predicting prognosis among patients with CKD.

## 1. Introduction

Chronic kidney disease (CKD) is a global health problem associated with a high risk of adverse clinical events and high health care costs [1]. In addition, CKD progression can induce the development of cardiovascular disease [2] and diabetes [3]. In the United States, Medicare costs associated with CKD and end-stage renal disease (ESRD) were reported to total more than $120 billion in 2017 [4]. The number of patients with ESRD is currently projected to increase to between 971,000 and 1,259,000 by 2030, representing a 41.3–83.2% increase in the prevalence from 687,093 patients with ESRD reported in 2015 [5]. In Taiwan, the hospitalization rate for ESRD has gradually increased from 964.1 per 1000 person-years in 2010 to 1037.9 per 1000 person-years in 2018 [6]. The total number of people on dialysis increased by 28.9%, from 65,610 patients in 2010 to 84,615 patients in 2018 [7]. The ESRD prevalence rate has gradually increased among older adults, especially among men 65 years and older.

CKD is typically silent and extremely variable. Moreover, the development of several chronic diseases have been associated with CKD progression, making clinical management particularly challenging. Timely interventions for patients with CKD could improve the quality of medical care and reduce morbidity, mortality, and healthcare costs [8,9]. Therefore, the development of a reliable model able to predict the risk of CKD progression even during early stages is necessary. Traditional statistical methods, such as the Cox hazard model, have been applied to the prediction of renal failure among patients with CKD in prior studies [10,11]. However, machine learning (ML) is increasingly being adopted to assess patients with CKD and consider all possible interactions between various input data [12,13,14]. A logistic regression (LR) classifier was able to predict the onset of renal replacement therapy (RRT) within 12 months with an area under the receiver operating characteristic curve (AUROC) value of 0.773 [15]. These results provided a screening approach for predicting the risk of RRT within 12 months. In another study, Subas et al. explored the abilities of several ML models for CKD diagnosis, including artificial neural network, support vector machine (SVM), k-nearest neighbor, C4.5 decision tree and random forest (RF) classifiers [16]. The RF model had the highest accuracy (100%), followed by the C4.5 decision tree classifier (99%), when applied to a dataset obtained from the University of California at Irvine ML Repository. Electronic health record (EHR) systems have drawn extensive and consistent attention and predictive models for clinical disease progression can be developed using features extracted from the EHR [17,18], allowing for the development and validation of predictive models that combine available laboratory data with data obtained from the EHR.

To provide a clinical database for medical assessments and improve healthcare quality, the pre-ESRD patient care and education program was initiated by the National Health Insurance (NHI) administration under the Ministry of Health and Welfare in Taiwan. The program includes CKD patients classified as Stages 3B to 5. In addition, the NHI reimbursement program has been shown to improve health care quality for patients with early-stage CKD (Stages 1 to 3A) [19]. The management program is required to follow specific clinical guidelines for each CKD stage. Patients with early-stage CKD are evaluated for urine protein to creatinine ratio (UPCR), serum creatinine levels, low-density lipoprotein cholesterol (LDL-C) levels and glycated hemoglobin (HbA1c) levels. Patients with pre-ESRD CKD are assessed for hemoglobin, blood urea nitrogen (BUN), serum creatinine levels, albumin levels, serum calcium levels, serum phosphate levels, fasting glucose levels, HbA1c levels, LDL-C levels, uric acid levels, sodium levels, potassium levels, triglyceride levels and UPCR.

In the present study, predictive models for progression from pre-dialysis CKD to ESRD were established for patients with Stages 2 to 5 CKD using various ML algorithms, including LR, RF, extreme gradient boosting (XGBoost), SVM and Gaussian naïve Bayes (GNB). Important classification features were evaluated for their value as high-risk factors to determine the optimal predictive features for use in each of the five models. Optimal model development could promote early CKD diagnosis and improve CKD management, preventing progression to kidney failure. These results could improve access to timely treatments among patients with CKD.

## 2. Material and Methods

### 2.1. Patient Population

In the retrospective cohort study, a total of 858 patients enrolled in the NHI program and diagnosed with early- (Stages 2 and 3A) or advanced-stage (Stages 3B, 4 and 5) CKD were treated from November 2006 to December 2019 at a branch of the Taipei Veterans General Hospital, including 516 with early-stage CKD and 342 with advanced-stage CKD. Early-stage CKD was defined as Stage 2 if 60 < eGFR < 89.9 mL/min/1.73 m^2^ and as Stage 3 if 45 < eGFR < 59.9 mL/min/1.73 m^2^. Advanced-stage CKD was defined as Stage 3B if 30 < eGFR < 44.9 mL/min/1.73 m^2^, as Stage 4 if 15 < eGFR < 29.9 mL/min/1.73 m^2^ and as Stage 5 if eGFR < 14.9 mL/min/1.73 m^2^. Based on the NHI program requirements, eGFR was calculated using the simplified Modification of Diet in Renal Disease equation. The outcome of this study was ESRD, defined as the diagnosis of renal failure, the initiation of hemodialysis or peritoneal dialysis. None of our study subjects were treated by kidney transplantation. Transfer to other hospitals, death and loss to follow-up were regarded as observation endpoints without reaching ESRD. De-identified data associated with patients diagnosed with Stages 2–5 CKD and enrolled in the two NHI CKD programs were retrieved from the hospital information database. The study was reviewed and approved by the Institutional Review Board (IRB) of Taipei Veterans General Hospital (No. 2020-01-024BC). Due to the use of de-identified data, the need for informed consent was waived by the IRB.

### 2.2. Study Design

In total, 858 patients with CKD were enrolled in this study, comprising 119 patients with Stage 2, 397 with Stage 3A, 111 with Stage 3B, 143 with Stage 4 and 88 with Stage 5. The numbers of early-stage CKD patients who progressed to ESRD within 3 and 5 years were 44 (5 with stage 2 and 39 with stage 3A) and 50 (6 with stage 2 and 44 with stage 3A), respectively. The numbers of advanced CKD patients who progressed to ESRD within 1 and 3 years were 38 (10 with stage 4 and 28 with stage 5) and 59 (2 with stage 3b, 17 with stage 4 and 40 with stage 5), respectively. A flow chart of the patient selection and categorization processes is shown in Figure 1.

The dataset was divided into patients managed with and without ESRD. In addition to the original dataset, the synthetic minority over-sampling technique (SMOTE) was also applied. Randomized data subsets were used for cross-validation (K = 5). The LR, RF, XGBoost, SVM and GNB classifiers were used to determine whether pre-dialysis CKD data could be used to predict progression to ESRD. The Shapley additive explanations (SHAP) value was used to select important characteristic factors for predicting CKD progression. Model retraining was performed using the most important risk factors for CKD progression to achieve optimal classification outcomes. The progression of CKD to kidney failure among patients diagnosed with early-stage CKD was followed for up to 5 years and the models were used to identify risk factors for CKD progression within 3 and 5 years. The progression of CKD to kidney failure in patients diagnosed with advanced-stage CKD was followed for up to 3 years and the models were used to identify risk factors for CKD progression within 1 and 3 years. The flow chart of model training and performance evaluation is shown in Figure 2.

### 2.3. Variables

Each subject’s predictor variables and baseline characteristics were obtained initially during the first clinic visit. The clinical characteristics of CKD were classified into four categories. Demographic variables included age, sex, height and weight. Laboratory data included serum and urine assessments, including eGFR, hemoglobin, hematocrit, creatinine, BUN, sodium, potassium, calcium, phosphorus and UPCR. Comorbid conditions included hypertension, diabetes and cardiovascular diseases. Risk-related biophysical and biochemical data included blood pressure, lipid profile and HbA1c levels. Those variables missing greater than 30% of values were excluded from the analysis. The missing values for other variables were replaced with multiple imputation. The study created five datasets using the multivariate imputation via chained equations module in the R package to perform the data imputation. All baseline characteristics and laboratory variables were obtained from the NHI pre-ESRD Patient Care and Education Program and the NHI Reimbursement Plans that Improve Health Care Quality of Early-Stage CKD Program implemented by the NHI.

### 2.4. Statistical Analysis

In this study, baseline demographic and laboratory data from the first clinic visit at which CKD was diagnosed were used to train the models. Due to differences in the collection of clinical characteristics, clinical data for Stages 2–3A CKD (early stage) were processed separately from clinical data for Stages 3B–5 CKD (advanced stage). Clinical indicators, including age, eGFR, serum creatinine, urine creatinine, LDL-C, HbA1c and UPCR, were associated with a significant risk of ESRD among patients diagnosed with Stages 2–5 CKD. In addition to these risk factors, the indicators associated with a significant risk for ESRD among Stages 3A–5 CKD include uric acid, albumin, fasting plasma glucose (FPG), triglyceride, cholesterol, hemoglobin, hematocrit, BUN, sodium, potassium, calcium and phosphorus. The detailed demographic characteristics of the cohort are listed in Table 1. Five available classification models were developed and the predictive performances of each model for determining the progression risk of various stages of pre-dialysis CKD were analyzed. The predictive performances of the five models were evaluated using an AUROC analysis and the sensitivity, specificity, accuracy, precision, F1 score and negative predictive value (NPV) were calculated. A cutoff value was identified based on the AUROC analysis to provide optimal sensitivity and specificity. Important features were selected and applied to determine the optimal model for predicting ESRD progression in patients with early- and advanced-stage CKD.

## 3. Results

The characteristics of patients with early-stage CKD (Stages 2 to 3A) and advanced-stage CKD (Stages 3B to 5) are shown in Table 1. The majority of patients were approximately 80 years old and most were men. High proportions of patients with CKD had comorbid diabetes and hypertension, particularly those with advanced-stage CKD. For example, the proportions of patients with hypertension and Stages 3B, 4 and 5 CKD were 75.7%, 74.1% and 79.5%, respectively. The proportions of patients with Stages 4 and 5 CKD undergoing ESRD were higher than those with Stages 2 and 3 CKD. Serum creatinine, urine creatinine, UPCR and HbA1c levels increased from Stages 2 to 5, whereas eGFR levels decreased as CKD progressed from early stages to advanced stages.

The predictive performances of the tested five models were evaluated using the AUROC analysis and other discrimination indicators. Significant performance differences were identified between the five models. The performances of each model to predict the progression of early- and advanced-stage CKD to ESRD are shown in Table 2 and Table 3, respectively. For early-stage CKD, the AUROC value for the RF classifier using SMOTE was 0.97 for predicting progression to ESRD within 3 years. Both RF and XGBoost classifiers with SMOTE resulted in AUROC values of 0.98 when predicting progression to ESRD within 5 years. For advanced-stage CKD, both RF and XGBoost classifiers with SMOTE resulted in AUROC values of 0.99 for the prediction of progression to ESRD within 1 year. The AUROC value of the RF classifier with SMOTE was 0.97 for the prediction of progression to ESRD within 3 years. The accuracy, specificity and sensitivity of the RF classifier were all greater than 90%. The AUROC plots of the RF classifier with SMOTE for predicting the progression of both early- and advanced-stage CKD are shown in Figure 3. The best performances among all five models were obtained using the RF classifier, which is suitable for predicting the progression of CKD to ESRD at all CKD stages.

To assess the contributions of various features to the prediction of ESRD progression, the SHAP value method was applied. For the progression of early-stage CKD to ESRD within 3 and 5 years, 13 features were analyzed by SHAP, as shown in Figure 4a,b. The results showed that urine creatinine and eGFR levels are the most influential features. A lower urine creatinine level is associated with a lower risk of progression to ESRD, whereas a lower eGFR level is associated with a higher risk of progression to ESRD. In addition, for the progression of advanced-stage CKD to ESRD within 1 and 3 years, 24 features were analyzed by SHAP, as shown in Figure 4c,d. The results indicated that serum creatinine level was the most important predictive feature in advanced-stage CKD. A lower serum creatinine level is associated with a lower risk of progression to ESRD. The second-most important features are hematocrit and urine creatinine, which have predictive value for progression within 1 and 3 years, respectively. A negative correlation between progression and hematocrit level was observed for progression within 1 year. Features associated with the progression of advanced-stage CKD to ESRD within 1 year showed more pronounced positive and negative associations with the risk of progression than the features associated with progression in 3 or 5 years for either advanced- or early-stage CKD.

According to the SHAP analysis, the top six features were used to retrain the model for predicting the progression of early-stage CKD within 5 years, as shown in Figure 5a,b. In addition, the top 10 features were used to retrain the model for predicting the progression of advanced-stage CKD within 1 year, as shown in Figure 5c,d. The AUROC values for the RF classifier with SMOTE were 0.96 for early-stage CKD and 0.97 for advanced-stage CKD. These results indicated that the RF classifier was the best model for predicting the risk of progression among patients with CKD, especially for predicting progression within 1 year in patients with advanced-stage CKD.

## 4. Discussion

In the present study, we evaluated several models for the ability to predict the progression of Stages 2–5 pre-dialysis CKD to ESRD. Based on our results, the SMOTE method can significantly improve the abilities of five models to predict CKD progression. The RF classifier showed the highest AUROC of 0.99 for predicting progression from advanced-stage CKD to ESRD within 1 year while achieving a sensitivity of 0.96, a specificity of 0.91, an accuracy of 0.93 and a precision of 0.90, as shown. Similarly, the XGBoost classifier showed an AUROC of 0.99, a sensitivity of 0.96, a specificity of 0.94, an accuracy of 0.95 and a precision of 0.93 for predicting the progression of advanced-stage CKD to ESRD within 1 year. Compared with the other classifiers, both the RF and XGBoost classifiers are more suitable for the early prediction of progression in patients with advanced-stage CKD.

In patients with early-stage CKD, the performance of the RF classifier was better than the performance of the XGBoost classifier, including better sensitivity, specificity, accuracy, precision and F1 scores. The AUROC values for the RF classifiers were 0.97 and 0.98 for predicting progression to ESRD within 3 and 5 years, respectively. Therefore, the RF classifier demonstrated the best performance for predicting progression in patients with both early- and advanced-stage pre-dialysis CKD. In a different study, Ravindra et al. reported an accuracy of 0.94 achieved by an SVM neural network for distinguishing between CKD and non-CKD [20]. In addition to classifier selection, feature selection is important for the performance of ML algorithms. Dulhare and Ayesha [21] indicated that the selection of suitable features or predictors is crucial for training ML classifiers. Their results showed that the GNB classifier had optimal performance when operated by a one-rule attribute selector.

In our study, we ranked the features associated with early- and advanced-stage CKD according to the SHAP value. The primary impact of urine creatinine can be observed for both early- and advanced-stage CKD. The results indicated that low urine creatinine levels are associated with a low risk of progression to ESRD, whereas eGFR and systolic blood pressure are risk factors for progression that can be observed during early-stage CKD. These results demonstrated that a low eGFR level is associated with a high risk of ESRD. High systolic blood pressure is an important risk factor that can be identified in early-stage CKD. Seyedzadeh et al. reported that ESRD was concurrent with several clinical symptoms, among which hypertension (52.3%) was the most commonly identified symptom in 128 patients [22]. In our study, a high prevalence of hypertension (79.5%) was observed among Stage 5 CKD patients. However, a negative association was identified between systolic blood pressure and ESRD in advanced-stage CKD based on the SHAP value. A high impact of serum creatinine was also identified for advanced-stage CKD. Therefore, in addition to serum creatinine, urine creatinine, eGFR and blood pressure are all highly associated with ESRD and each factor has different impacts at different CKD stages. Urine creatinine appears as a parameter of special relevance to predict the evolution of kidney damage. Nonetheless, it should be considered that the parameter may be altered due to urine dilution in different time of sample collection. Our study suggests that multiple parameters are needed simultaneously while using the prediction model. The calculation of eGFR has been offered as a practical and easy approach for converting serum creatinine values, as reviewed by Mula-Abed [23]. Further study of the relationship between eGFR and creatine will improve the qualitative estimation of the interaction between eGFR and creatinine and their impacts on ESRD in CKD patients. Additionally, our results showed that high serum phosphate is a common complication in patients with ESRD. Seyedzadeh et al. reported a large impact for serum phosphate in approximately 50% of patients with ESRD, associated with renal osteodystrophy [22]. Our study found increased serum phosphate levels in patients with Stages 3B to 5 CKD, which is consistent with the previous study.

Anemia or low serum albumin are also common complications of CKD. Our findings indicated that low hematocrit and albumin levels are associated with an increased risk of ESRD among patients with advanced-stage CKD. Anemia is strongly associated with poor kidney function in CKD patients [24]. A previous study reported prevalence rates of anemia among patients with CKD of 42%, 33%, 48%, 71% and 82% for Stages 1, 2, 3, 4 and 5, respectively, in Saudi Arabia [25]. Decreased serum albumin levels were also associated with a decline in eGFR and may be related to proteinuria or underlying inflammation [26]. The criteria for hyperuricemia include an increased uric acid level, which was also observed to increase from Stages 3B to 5 in the current study. These results implied that high uric acid levels in patients with advanced-stage CKD might be associated with impending renal failure. Past work indicated a ‘J-shaped’ association between uric acid levels and mortality in hemodialysis patients [27]. Thus, maintaining uric acid levels in patients with advanced-stage CKD within normal levels should be a clinical goal and uric acid levels should be monitored. Oda and Kawai reported that LDL-C levels were significantly higher in Stages 2 and 3 CKD than in Stage 1 CKD in a study including 3897 patients [28]. In our study, the LDL-C levels in patients with Stages 3B and 4 CKD were higher than those in patients with early-stage CKD, which is consistent with the findings of the previous study and provides additional information for more advanced CKD stages. Due to collinearity and the reduced impacts of hypertension and diabetes in our models, these two comorbidities were excluded from our models. HbA1c levels are a well-known indicator of diabetes control and showed a positive impact on the progression of advanced-stage CKD. Based on our results, the impacts of risk factors differ across different stages of CKD. Thus, specific management strategies are necessary for different stages, making the early diagnosis of CKD particularly important.

To assess the optimal predictive model for the progression of pre-dialysis CKD to ESRD, the top six features (eGFR, blood pressure, UPCR, serum creatinine, urine creatinine and LDL-C) identified for early-stage CKD and the top 10 features (serum creatinine, uric acid, urine creatinine, calcium, LDL-C, hemoglobin, HbA1c, cholesterol, phosphorus and triglyceride) identified for advanced-stage CKD were used to retrain all five models. The RF classifier demonstrated the optimal risk prediction performance for the progression of pre-dialysis CKD to ESRD. The AUROC values for the RF classifier with SMOTE were 0.96 for the progression of early-stage CKD within 5 years and 0.97 for the progression of advanced-stage CKD within 1 year. A slight difference (0.98 to 0.96) was observed for the RF classifier when using only six features compared with using all features for the prediction of early-stage CKD progression. Similarly, the AUROC of the RF classifier using only the top 10 features showed a slight decline (0.99 to 0.97) compared with the RF classifier using all features when predicting advanced-stage CKD progression. The results indicated that the RF classifier could be used with specific features to predict ESRD risk in patients with pre-dialysis CKD. This approach can help clinicians understand the risk factors associated with ESRD and the progression of patients with CKD at different stages.

An effective predictive model can help medical teams quickly and easily identify the key factors contributing to the deterioration of renal function, track the rate of renal function decline and modify the care goals on a rolling basis. In addition, predicting the time of progressing to ESRD can early remind care providers, patients and relatives with facing to the dangers and complications of ESRD. Certain strategies, such as stricter diet control, treatment of electrolyte imbalances and acidemia, improvement of anemia and uremia, or early decision on dialysis mode can be intervened in time to reduce the impact on the body and on life. 

Our approach provides a reference to clinical strategy. Nonetheless, there are several limitations to this study. First, this cohort consisted of a relatively small sample, so the model performance may have been affected by the training data. Second, the laboratory data was limited by geography and subject demographics, limiting the generalizability of these findings to the wider population. Third, the individual laboratory records may have changed over time. Because our models were based on baseline data, our study cannot present the trajectory of disease progression. A longitudinal model is likely to better reflect associations between risk factors and disease progression and should be evaluated in future studies

## 5. Conclusions

The present study demonstrated a reliable ML method for predicting the risk of progression to eventual ESRD among patients with Stages 2–5 CKD. The RF classifier with SMOTE showed the best performance for the early diagnosis of CKD prognosis. In addition, the high performance of ML classifiers was achieved when limiting the analysis to predominant features. This approach reveals that the RF classifier is suitable for risk assessments among patients with pre-dialysis CKD and the results could be potentially advantageous for patient screening initiatives.

## Figures and Tables

**Figure 1 diagnostics-12-02454-f001:**
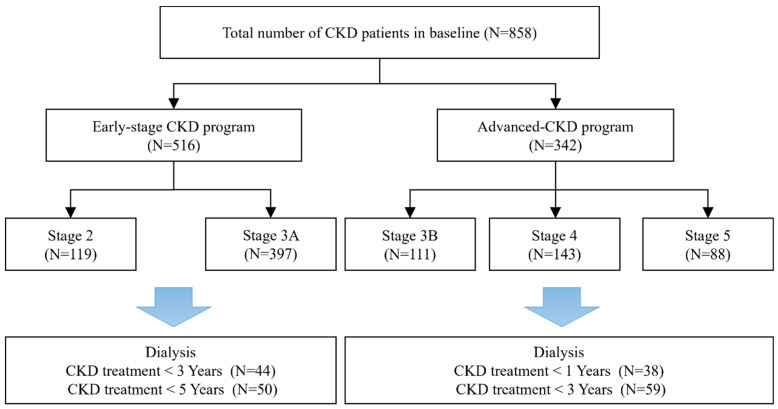
Flow chart for the selection of study subjects.

**Figure 2 diagnostics-12-02454-f002:**
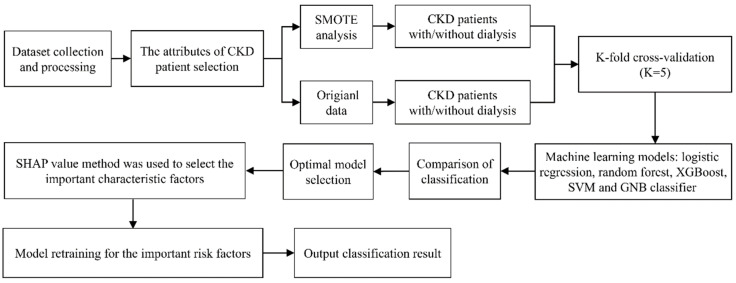
Flow chart of model training and performance evaluation.

**Figure 3 diagnostics-12-02454-f003:**
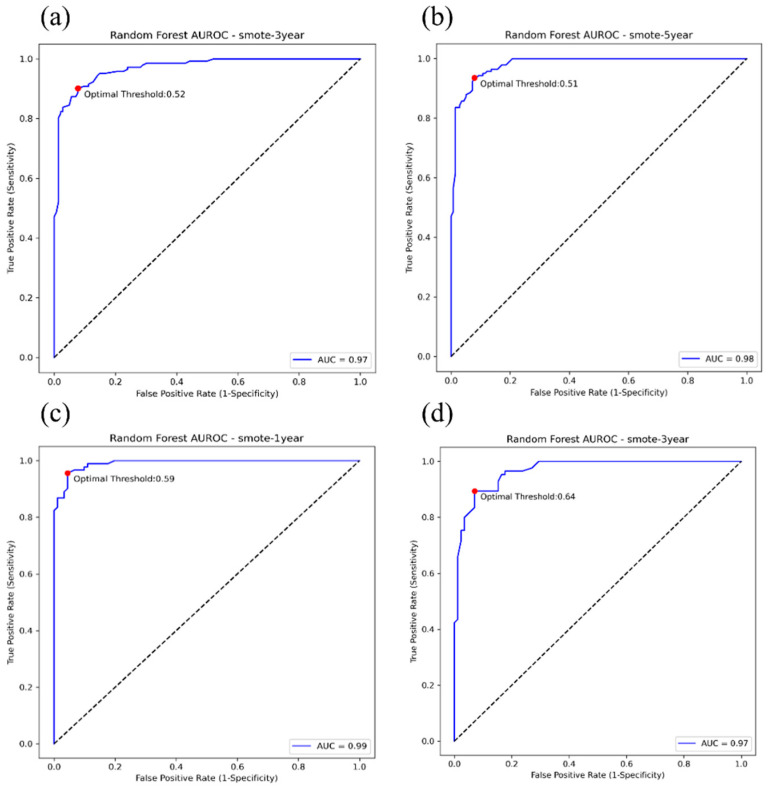
Plots showing the area under the receiver operating characteristic curve (AUROC) for the ability of the random forest (RF) classifier trained using synthetic minority over-sampling technique (SMOTE) to predict the progression of early-stage chronic kidney disease (CKD) to end-stage renal disease (ESRD) within (**a**) 3 and (**b**) 5 years. Plots showing the AUROC for the ability of the RF classifier trained using to predict the progression of advanced-stage CKD to ESRD within (**c**) 1 and (**d**) 3 years.

**Figure 4 diagnostics-12-02454-f004:**
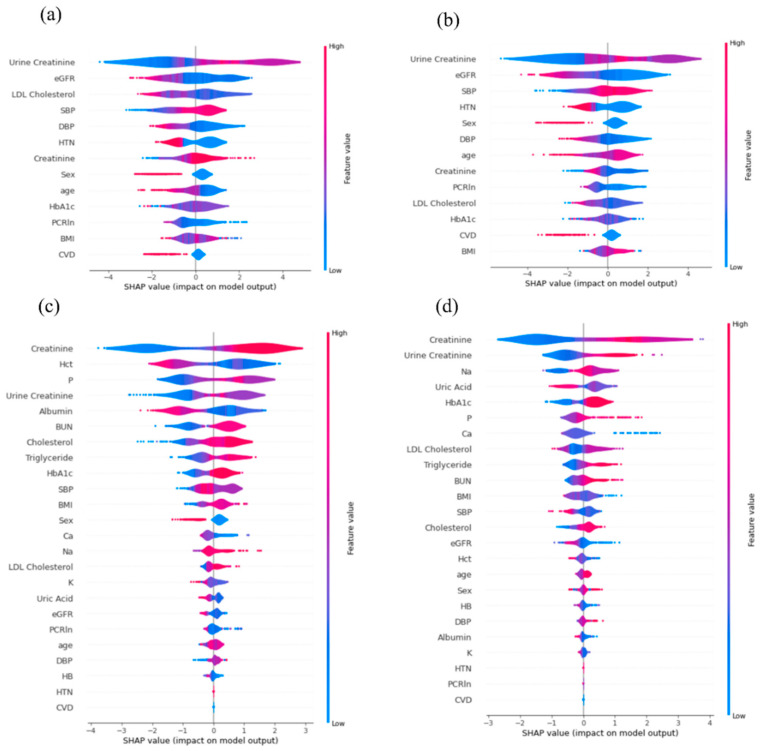
Positive and negative impacts of 13 features on the prediction of progression from early-stage chronic kidney disease (CKD) to end-stage renal disease (ESRD) within (**a**) 3 and (**b**) 5 years. Positive and negative impacts of 24 features on the prediction of progression from advanced-stage CKD to ESRD within (**c**) 1 and (**d**) 3 years. Features are ranked in descending according to Shapley additive explanations (SHAP values), where the top feature represents the most informative feature. Each dot in the plot represents the value for an individual patient. The color represents the scale of the feature’s value, ranging from high (red) to low (blue), for the observation.

**Figure 5 diagnostics-12-02454-f005:**
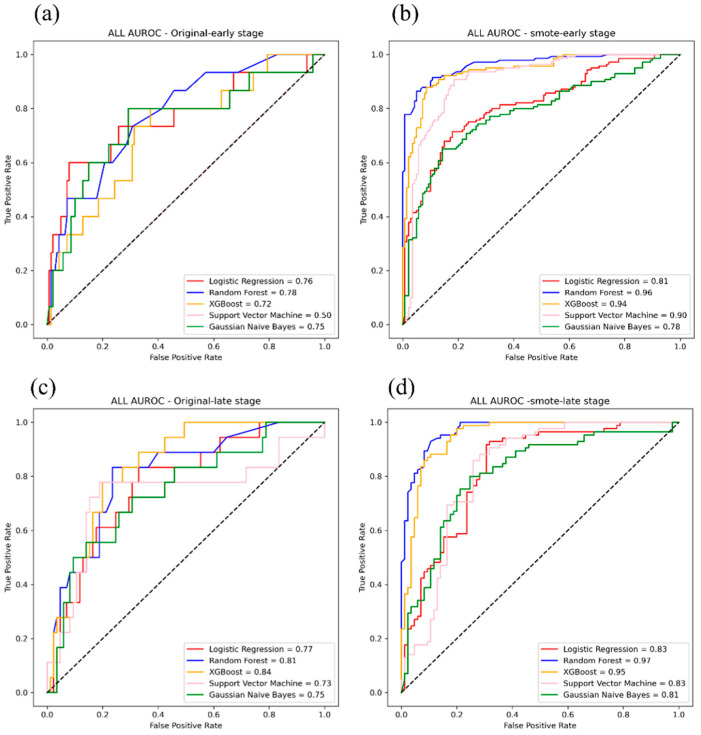
Plots showing the under the receiver operatic characteristic curve (AUROC) of five models, including six features from (**a**) original and (**b**) synthetic minority over-sampling technique (SMOTE) data for predicting end-stage renal disease progression within 5 years in patients with early-stage chronic kidney disease. Plots showing the AUROC of five models, including 10 features from (**c**) original and (**d**) SMOTE data for predicting end-stage renal disease progression within 1 year in patients with advanced-stage chronic kidney disease. GNB, Gaussian naïve Bayes; LR, linear regression; RF, random forest; SVM, support vector machine; XGBoost, extreme gradient boosting.

**Table 1 diagnostics-12-02454-t001:** Characteristics of CKD patients according to disease stage.

	Cases (Early-Stage CKD Patients) N = 516	Stage 2 N = 119	Stage 3A N = 397	Cases (Advanced-Stage CKD patients) N = 342	Stage 3B N = 111	Stage 4 N = 143	Stage 5 N = 88
Age (years)	80.8 ± 11.5	79.8 ± 11.2	81.1 ± 11.5	79.4 ± 0.7	79.2 ± 1.1	80 ± 1.2	78.5 ± 1.4
Sex (%)							
Male	359 (70%)	84 (71%)	275 (69%)	227 (66.4%)	73 (65.8%)	103 (72%)	51 (58%)
Female	157 (30%)	35 (29%)	122 (31%)	115 (33.6%)	38 (34.2%)	40 (28%)	37 (42%)
Cardiovascular diseases (%)							
None	214 (41%)	58 (49%)	156 (39%)	318 (93%)	102 (91.9%)	132 (92.3%)	84 (95.5%)
Yes	302 (59%)	61 (51%)	241 (61%)	24 (7%)	9 (8.1%)	11 (7.7%)	4 (4.5%)
Diabetes (%)							
None	242 (47%)	59 (50%)	183 (46%)	159 (46.5%)	59 (53.2%)	59 (41.3%)	41 (46.6%)
Yes	274 (53%)	60 (50%)	214 (54%)	183 (53.5%)	52 (46.8%)	84 (58.7%)	47 (53.4%)
Hypertension (%)							
None	398 (77%)	89 (75%)	309 (78%)	82 (24%)	27 (24.3%)	37 (25.9%)	18 (20.5%)
Yes	118 (23%)	30 (25%)	88 (22%)	260 (76%)	84 (75.7%)	106 (74.1%)	70 (79.5%)
Hemodialysis (%)							
None	466 (90%)	113 (95%)	353 (89%)	283 (82.7%)	109 (98.2%)	126 (88.1%)	48 (54.5%)
Yes	50 (10%)	6 (5%)	44 (11%)	59 (17.3%)	2 (1.8%)	17 (11.9%)	40 (45.5%)
BMI (kg/m^2^)	26.1 ± 12.6	26.9 ± 13.3	25.9 ± 12.3	24.6 ± 0.2	25.2 ± 0.4	24.8 ± 0.3	23.5 ± 0.5
Systolic blood pressure (mmHg)	133.4 ± 18.2	132.5 ± 18.3	133.6 ± 18.2	139.1 ± 1.2	137.3 ± 0.7	139.8 ± 1.8	140.4 ± 2.3
Diastolic blood pressure (mmHg)	72.6 ± 10.9	72.3 ± 11.5	72.6 ± 10.7	73.8 ± 0.9	73.6 ± 1.4	74.1 ± 1.2	73.5 ± 1.5
eGFR (mL/min/1.73 m^2^)	53.3 ± 11	68.4 ± 8.6	48.8 ± 6.9	24.3 ± 0.2	37.1 ± 0.4	23.1 ± 0.4	10.3 ± 0.4
Creatinine (mg/dL)	1.3 ± 0.3	1.0 ± 0.2	1.4 ± 0.3	3.3 ± 0.2	1.8 ± 0.1	2.6 ± 0.1	6.1 ± 0.4
Urine creatinine (mg/dL)	419.1 ± 717.4	309.9 ± 534.7	451.8 ± 760.7	1973.2 ± 168.9	987.2 ± 169.2	2067.7 ± 307.6	3063.4 ± 331.8
LDL-C (mg/dL)	101.8 ± 27.7	100.1 ± 26.5	102.3 ± 28	103.2 ± 1.8	102.6 ± 2.8	105.3 ± 3	100.3 ± 3.9
HbA1c (%)	6.6 ± 1.4	6.4 ± 1.2	6.6 ± 1.5	7.1 ± 0.3	7.03 ± 0.5	6.7 ± 0.1	7.7 ± 0.9
UPCR (mg/g)	5.3 ± 1.3	5.1 ± 1.5	5.4 ± 1.3	6.6 ± 0.1	5.9 ± 0.2	6.6 ± 0.2	7.4 ± 0.2
Uric Acid (mg/dL)				7.3 ± 0.11	6.8 ± 0.17	7.3 ± 0.16	7.8 ± 0.24
Albumin (g/dL)				3.6 ± 0.03	3.8 ± 0.04	3.6 ± 0.04	3.3 ± 0.06
FPG (mg/dL)				118.8 ± 2.5	102.6 ± 2.8	119.1 ± 4.4	116.6 ± 4.3
Triglyceride (mg/dL)				142.8 ± 4.8	149.7 ± 8.6	142.6 ± 0.7.1	134.4 ± 9.7
Cholesterol (mg/dL)				178.6 ± 2.5	177.1 ± 3.6	181.9 ± 3.9	175.3 ± 5,4
Hemoglobin (gm/dL)				11.1 ± 0.1	12.3 ± 0.2	11.2 ± 0.2	9.4 ± 0.2
Hematocrit (%)				33.8 ± 0.3	37.1 ± 0.6	34.1 ± 0.4	29.0 ± 0.5
BUN (mg/dL)				47.0 ± 1.4	29.9 ± 0.8	42.4 ± 1.3	76.1 ± 3.3
Sodium (Na) (mg/dL)				140.2 ± 0.2	140.6 ± 0.3	139.3 ± 0.3	140.1 ± 0.5
Potassium (K) (mg/dL)				4.6 ± 0.1	4.5 ± 0.1	4.7 ± 0.1	4.7 ± 0.1
Calcium (Ca) (mg/dL)				8.9 ± 0.1	9.3 ± 0.3	8.9 ± 0.05	8.5 ± 0.1
Phosphorus (P) (mg/dL)				4.1 ± 0.1	3.7 ± 0.1	4.0 ± 0.1	5.0 ± 0.2

BMI, body mass index; BUN, blood urea nitrogen; CKD, chronic kidney disease; eGFR, estimated glomerular filtration rate; FPG, fasting plasma glucose; HbA1c, glycated hemoglobin; LDL-C, low-density lipoprotein cholesterol; UPCR, urine protein and creatinine ratio.

**Table 2 diagnostics-12-02454-t002:** The performances of predictive models for the progression of early-stage CKD to ESRD.

Algorithm	Year	Dataset	Sensitivity	Specificity	Accuracy	Precision	F1-Score	NPV	AUROC
LR	3	Original	0.92	0.72	0.74	0.23	0.37	0.99	0.83
		SMOTE	0.79	0.87	0.83	0.85	0.82	0.80	0.90
	5	Original	0.73	0.78	0.77	0.26	0.39	0.96	0.79
		SMOTE	0.76	0.89	0.83	0.88	0.82	0.79	0.90
RF	3	Original	0.92	0.33	0.90	0.99	0.95	0.61	0.73
		SMOTE	0.90	0.94	0.91	0.94	0.91	0.88	0.97
	5	Original	0.96	0.30	0.81	0.81	0.88	0.73	0.79
		SMOTE	0.93	0.92	0.93	0.92	0.93	0.94	0.98
XGBoost	3	Original	0.95	0.21	0.79	0.81	0.87	0.54	0.65
		SMOTE	0.90	0.92	0.86	0.85	0.88	0.92	0.96
	5	Original	0.96	0.29	0.80	0.81	0.88	0.73	0.72
		SMOTE	0.91	0.91	0.91	0.91	0.91	0.91	0.98
SVM	3	Original	1.00	0.00	0.92	1.00	0.96	0.00	0.50
		SMOTE	0.85	0.85	0.85	0.85	0.85	0.85	0.92
	5	Original	0.90	0.00	0.90	1.00	0.95	0.00	0.50
		SMOTE	0.90	0.90	0.90	0.90	0.90	0.91	0.93
GNB	3	Original	0.95	0.43	0.63	0.61	0.75	0.85	0.80
		SMOTE	0.70	0.88	0.75	0.92	0.79	0.59	0.84
	5	Original	0.94	0.39	0.88	0.92	0.93	0.47	0.75
		SMOTE	0.76	0.79	0.78	0.80	0.78	0.74	0.84

AUROC, area under the receiver operating characteristic curve; CKD, chronic kidney disease; ESRD, end-stage renal disease; GNB, Gaussian naïve Bayes; LR, linear regression; NPV, negative predictive value; RF, random forest; SVM, support vector machine; SMOTE, synthetic minority over-sampling technique; XGBoost, extreme gradient boosting.

**Table 3 diagnostics-12-02454-t003:** The performances of predictive models for the progression of advanced-stage CKD to ESRD.

Algorithm	Year	Dataset	Sensitivity	Specificity	Accuracy	Precision	F1-Score	NPV	AUROC
LR	1	Original	0.90	0.69	0.72	0.26	0.41	0.98	0.84
		SMOTE	0.98	0.81	0.90	0.84	0.90	0.97	0.92
	3	Original	0.89	0.74	0.77	0.42	0.58	0.97	0.85
		SMOTE	0.88	0.72	0.80	0.76	0.82	0.86	0.85
RF	1	Original	0.99	0.42	0.85	0.85	0.91	0.91	0.92
		SMOTE	0.96	0.91	0.93	0.90	0.93	0.97	0.99
	3	Original	0.98	0.41	0.76	0.72	0.83	0.94	0.87
		SMOTE	0.90	0.93	0.91	0.93	0.91	0.89	0.97
XGBoost	1	Original	0.99	0.37	0.82	0.81	0.89	0.91	0.92
		SMOTE	0.96	0.94	0.95	0.93	0.94	0.96	0.99
	3	Original	0.94	0.41	0.77	0.76	0.84	0.78	0.82
		SMOTE	0.92	0.87	0.89	0.86	0.89	0.93	0.95
SVM	1	Original	0.94	0.38	0.85	0.89	0.92	0.55	0.63
		SMOTE	0.88	0.94	0.91	0.95	0.91	0.89	0.94
	3	Original	0.94	0.52	0.83	0.86	0.90	0.72	0.79
		SMOTE	0.96	0.84	0.89	0.81	0.88	0.96	0.92
GNB	1	Original	1.00	0.29	0.73	0.70	0.83	1.00	0.85
		SMOTE	0.92	0.86	0.88	0.85	0.88	0.92	0.92
	3	Original	0.97	0.41	0.75	0.73	0.83	0.89	0.85
		SMOTE	0.85	0.74	0.78	0.68	0.76	0.88	0.83

AUROC, area under the receiver operating characteristic curve; CKD, chronic kidney disease; ESRD, end-stage renal disease; GNB, Gaussian naïve Bayes; LR, linear regression; NPV, negative predictive value; RF, random forest; SVM, support vector machine; SMOTE, synthetic minority over-sampling technique; XGBoost, extreme gradient boosting.

## Data Availability

The datasets generated and analyzed during the current study are not publicly available due to privacy/ethical restrictions but are available from the corresponding author on reasonable request.

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
