# Peer review of "Machine Learning Models for the Prediction of Renal Failure in Chronic Kidney Disease: A Retrospective Cohort Study"

_diagnostics, 2022, doi:10.3390/diagnostics12102454_

Round 1

Reviewer 1 Report

This is an interesting study, since the prediction models that allow anticipating the progress of diseases are currently being valued. In this case, it is proposed to evaluate the usefulness of different machine learning models in the context of chronic kidney damage.

Some comments

- Figure 1. When patients who go to dialysis are mentioned, it would be interesting to know what stage they are referring to.

- Figure 4: urinary creatinine appears as a parameter of special relevance to predict the evolution of kidney damage, however it is not a parameter considered important when evaluating kidney function since, historically, it has been considered an indicator of concentration or urine dilution and is highly variable depending on the time of sample collection. Do you think it could be a promising parameter to identify the patients most susceptible to kidney damage?

Some comment is missing that values ​​the usefulness of these prediction models. What is the use of predicting that a patient will require dialysis when there is no strategy to avoid it?

Author Response

Dear reviewers,

    We appreciate the reviewer for the valuable and constructive comments on our paper. The changes made to the paper (please see the attachment) in response to the comments are summarized as follows.

Reviewer 1

1. Figure 1. When patients who go to dialysis are mentioned, it would be interesting to know what stage they are referring to.

Reply:

The numbers of early-stage CKD patients who progressed to ESRD within 3 and 5 years were 44 (5 with stage 2 and 39 with stage 3A) and 50 (6 with stage 2 and 44 with stage 3A), respectively. The numbers of advanced CKD patients who progressed to ESRD within 1 and 3 years were 38 (10 with stage 4 and 28 with stage 5) and 59 (2 with stage 3b, 17 with stage 4 and 40 with stage 5), respectively. We add the description in the revised version.

2. Figure 4: urinary creatinine appears as a parameter of special relevance to predict the evolution of kidney damage, however it is not a parameter considered important when evaluating kidney function since, historically, it has been considered an indicator of concentration or urine dilution and is highly variable depending on the time of sample collection. Do you think it could be a promising parameter to identify the patients most susceptible to kidney damage?

Reply:

We agree with your comment. Urine creatinine appears as a parameter of special relevance to predict the evolution of kidney damage. Nonetheless, it should be considered that the parameter may be altered due to urine dilution in different time of sample collection. Our study suggests multiple parameters is needed simultaneously while using the prediction model. The statement has been added in the Discussion section.

3. Some comment is missing that values the usefulness of these prediction models. What is the use of predicting that a patient will require dialysis when there is no strategy to avoid it?

Reply:

An effective predictive model can help medical teams quickly and easily identify the key factors contributing to the deterioration of renal function, track the rate of renal function decline, and modify the care goals on a rolling basis. In addition, predicting the time of progressing to ESRD can early remind care providers, patients and relatives with facing to the dangers and complications of ESRD. Certain strategies such as stricter diet control, treatment of electrolyte imbalances and acidemia, improvement of anemia and uremia, or early decision on dialysis mode can be intervened in time to reduce the impact on the body and life.

Reviewer 2 Report

Su et al. reported the use of machine learning models to predict early and advanced CKD progression by using baseline laboratory data and demographics in the models. They found that the RF classifier method with SMOTE had the best accuracy and suggested that this could be used as a screening tool for CKD patients.

This work is highly interesting. All of the major limitations have been alluded to by the authors, namely the lack of longitudinal data. It is quite surprising that using cross-sectional data at baseline could predict ESRD for up to 5 years.

Minor comments:

1. Were there any missing data? How did the authors deal with missing data?

2. Apart from retraining the models, were the models externally validated?

3. Was eGFR included in the model for late CKD stage? If patients had CKD stage 5, it is likely that low eGFR would trigger RRT initiation and indicate bias from treating physicians.

Author Response

Dear reviewers,

    We appreciate the reviewer for the valuable and constructive comments on our paper. The changes made to the paper (please see the attachment) in response to the comments are summarized as follows.

Reviewer 2:

1. Were there any missing data? How did the authors deal with missing data?

Reply:

Those variables missing greater than 30% of values were excluded from the analysis. The missing values for other variables were replaced with multiple imputation. The study created five datasets using the multivariate imputation via chained equations (MICE) module in the R package to perform the data imputation. The statement has been added in the Discussion section.

2. Apart from retraining the models, were the models externally validated?

Reply:

Because of the difficulty to access external sample, our models did not run an external validation.

3. Was eGFR included in the model for late CKD stage? If patients had CKD stage 5, it is likely that low eGFR would trigger RRT initiation and indicate bias from treating physicians.

Reply:

We agree with your comment. If patients had CKD stage 5, it is likely that low eGFR would trigger RRT initiation. eGFR was included in the model for advanced-stage CKD initially. However, it was excluded in the retrain model. Irrespective to eGFR, the performance of the models is not different much.